# An Extensive Analysis of the Engineering Design of Underground Sewage Plants in China

Abdulmoseen Segun Giwa [1] and Nasir Ali [2,*]

1   School of Environment and Civil Engineering, Nanchang Institute of Science and Technology,
    Nanchang 330108, China; giwasegun@live.com
2   Institute of Biotechnology Genetic Engineering, The University of Agriculture,
    Peshawar 25130, Khyber Pakhtunkhwa, Pakistan
*   Correspondence: nasir.biotech@yahoo.com

**Abstract:** In recent years, underground sewage treatment plants that can remarkably reduce land occupation with less environmental pollution are gradually entering the popular consciousness and are now being used widely. However, problems associated with the traditional treatment plants, such as high construction and operation cost, severe health and safety risks, and monotonous landscape design have limited their value and restricted their application and promotion. Through the literature and field investigations, the value of underground sewage treatment plants (STPs) was analyzed, their engineering and landscape design were studied, and their development direction was explored in order to supply a theoretical basis for further application and development of underground STPs. The analysis showed that as a new model of environmentally friendly sewage treatment and resource conservation, underground STPs have the apparent advantages of lower cost of land use for construction and pipeline as well as an outstanding value for the urban landscape and ecological environment. These factors can offset its relatively high construction and operating costs to a certain extent, especially when compared with above-ground STPs. The engineering design study results showed that significant differences existed between underground STPs and traditional above-ground STPs, and that the main contents of the engineering design of underground STPs consist of treatment scale and degree, influent and effluent qualities, site selection, design model, underground arrangement and structure, main treatment process, monomer structure, ventilation and deodorization, daylighting and artificial lightings, fire safety, operation and maintenance, and the linkage design between the above-ground landscape and the underground STP.

**Keywords:** engineering design; landscape design; underground sewage treatment environmental pollutions

## 1. Introduction

With reduced available land area and the increasingly prominent problems of environmental pollution, construction of underground sewage treatment plants that occupy less space, save land resources, cause less environmental pollution, and that can coordinate with the surrounding environment will gradually become a hot spot in industries and become a new development trend of large-scale urban sewage treatment projects [1]. The applicable conditions and scope of an underground sewage treatment plant include small amounts of sewage, less available land on the ground, special requirements for the above-ground landscape, low groundwater level, an easy-to-excavate geology, and an adequate burying depth of the sewage return pipe [1,2]. In addition, the approval and construction of an underground sewage treatment plant rely on it being at a certain distance from residential buildings; it should not be located in a depression area or in an area accumulating rainwater, and further there is a need for less strict local sewage discharge requirements in the areas. Generally speaking, underground sewage treatment plants are suitable for construction in

areas where thermal insulation needs to be considered, where there is less requisitionable land area, and where there are higher environmental requirements [3–5].

However, major problems such as ventilation, deodorization, and hidden safety risks as well as issues like flooding during the rainy season, rising effluent, sludge digestion, and the inability to reserve space for upgrading and transformation bring challenges to the construction of underground sewage treatment plants [6]. Attention must be paid to factors such as the limitation of land resources in China. In addition, the gradual improvement in effluent quality standards also restricts the design and application of underground sewage treatment plants to a certain extent. It is believed that, except for the design scale and water quality, which are not significantly different from those of traditional above-ground sewage treatment plants, there are at least 10 aspects of process design in underground sewage treatment plants that are different from above-ground sewage treatment plants. These include structural layout, process optimization, energy saving and consumption reduction, ventilation and deodorization, lighting and daylighting, fire safety, operation and maintenance, maintenance and technical transformation, flood control and drainage, and emergency treatment technology. It is necessary to have core technologies that match those of the above-ground sewage treatment plants in terms of pretreatment, biological treatment, and post-treatment enhancement. Based on relevant research and reports, there are corresponding norms and standards for underground sewage treatment plants in the United States, Japan, and the European Union. However, in China, due to the delayed start of research and development of underground sewage treatment plants, there are still no unified technical standards or norms to guide the engineering design and operation of underground sewage treatment plants.

In view of the existing problems in the above-ground sewage treatment plants, researchers have to tried to change the design and construction ideas of sewage treatment plants. In this context, construction of an underground sewage treatment plant with small footprint, space-saving, and low environmental pollution features has gradually entered the popular consciousness. Such a plant can solve the problem of the impact of noise and odor generated during sewage treatment on the lives of residents around the water plant; it also allows sewage to be treated locally, does not require a long-distance pipe network, does not affect the urban environment, and can also beautify the urban environment, all of which meet the needs of urban ecological civilization construction and development. A comparison effect of above-ground and underground sewage treatment plants is shown in Figure 1.

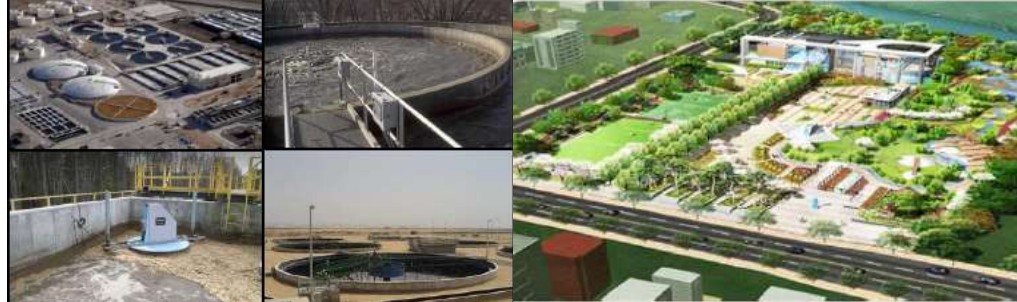

**Figure 1.** Comparison of landscape effects between traditional above-ground sewage treatment plants and underground sewage treatment plants.

This paper introduces the technical points and characteristics of the engineering design of underground sewage treatment plants based on a systematic analysis of the current problems in these plants. It explains the existing problems and suggests countermeasures, puts forward a set of suitable steps for guiding the engineering and technical implementation of the construction plan, operation, maintenance and management of underground sewage treatment plants in China, and provides a technical reference for the compilation of China's "engineering technical specifications for underground sewage treatment plants".

## 2. Purpose and Objectives

"Water Pollution Prevention and Control Action Plan" (also known as "Water Ten Measures") was officially promulgated in 2015, which pointed out the direction for China's "13th Five-Year Plan" period for water pollution control. As the top priority of China's water environment control, the treatment of urban domestic sewage will also be greatly developed. Urban domestic sewage treatment will not only pay attention to the treatment cost and effluent quality in the future but also be mindful toward the socio-economic and environmental benefits generated by sewage treatment. Therefore, in the future, underground sewage treatment plants with such significant advantages will surely become a new choice and development trend for large-scale urban sewage treatment projects in China.

This paper intends to conduct a research on the design formation and main processes of optimization selection of underground sewage treatment plants, engineering structure design, and ground landscape design in view of the many problems persisting in the construction and application of underground sewage treatment plants in China. It also analyzes and discusses the key issues in the development and application of underground sewage treatment plants suitable for China's national conditions, and provides technical support and a theoretical basis for the in-depth promotion and wide application of underground sewage treatment plants in China in the future.

This article combines the literature and the field research of some underground sewage treatment plants operating in China and intends to carry out relevant analysis from the following aspects.

1. Value analysis

On-site investigations of some of the underground sewage treatment plants in operation in China are conducted to understand and grasp the engineering investment, effluent water quality, and social, economic and environmental benefits of different design forms and different treatment processes of underground sewage treatment plants. The advantages and disadvantages of different design forms and main processes of underground sewage treatment plants in terms of cost and benefit are compared and analyzed, and the optimal design form and process of underground sewage treatment suitable for different treatment scales and effluent quality requirements are proposed according to local conditions.

2. Engineering design

We study the engineering design to guide the construction and operation of underground sewage treatment plants in view of the fact that the design and construction of these plants are significantly different from those of the above-ground sewage treatment plants in China. The design of treatment scale and degree of treatment, project site selection and construction form design, underground layout and structure design, main process design, single structure design, ventilation, deodorization design, daylighting and lighting design, fire protection and safety design, operation, and maintenance design are certain preferences that could describe the significance of underground sewage treatment plants as compared with above-ground sewage treatment plants.

In view of the relatively backward design concept, relatively single model, and insufficient functions in the above-ground landscape design of underground sewage treatment plants in China, some key contents of landscape design such as design style, design features, design concept, eco-friendliness, and maintenance costs are discussed. Combined with actual engineering cases, the cost and benefit of above-ground landscape design are analyzed, proposals for above-ground landscape design of underground sewage treatment plants are put forth, and the value and significance of above-ground landscape design for underground sewage treatment plants are pointed out.

3. Model and technology development direction

Aiming at the current problems of China's underground sewage treatment plants, such as single construction model, insufficient landscape design innovation, and narrow

development ideas for above-ground space, we discuss the breakthroughs in key sewage treatment technologies, diversification of investment and operation methods, and large-scale promotion of semi-underground underground sewage treatment plants from the from the perspective of design concept innovation. The key contents of the development and application of underground sewage treatment plants for the future is preliminarily discussed in order to further improve the social, economic, and environmental benefits of underground sewage treatment plants and prevent underground pollution. It provides a reference for the construction cost of traditional sewage treatment plants and their promotion and application.

In view of the different design forms of sewage treatment plants such as underground, semi-underground, above-ground, above-ground fully enclosed, and different sewage treatment processes such as $A^2O$, MBR, and SBR, which are currently used in urban sewage treatment plants in China, this section lists the relevant research. It outlines the treatment scale, service area, overall investment, design form, secondary and main processes of advanced treatment, influent and effluent water quality, and engineering characteristics of each sewage treatment plant, providing a basis for subsequent comparative analysis.

*2.1. Fully Underground*

1.    Shenzhen Buji Sewage Treatment Plant—$A^2O$-Biofilm Co-pool Process

The Shenzhen Buji Sewage Treatment Plant is located in Buji Street, Shenzhen. Its construction began in April 2008 and its trial operation commenced in April 2011. The construction period was nearly 3 years. The designed sewage treatment's capacity is 200,000 tons/day. The load can reach 260,000 tons/day. The planned land area of the project is 5.95 hectares. After excluding the area occupied by river improvement and diversion, the actual land area is 4.6 hectares, the total investment is CNY 590 million, the service population is 450,000, and the sewage discharge implements a first-class model.

The planned land of the Buji Sewage Treatment Plant is narrow, long, and irregular. An underground design is adopted in order to effectively save land resources and improve land use efficiency. The main structures are all located underground, and the upper space is built as a municipal leisure park. The construction area of the leisure park is about 4.30 hectares. The thickness of the top layer of the underground structure is 1.5 m and deep-rooted large tropical plants are planted therein. The overall construction of the factory environment and leisure park landscape can not only coordinate with the surrounding topography but also be integrate with the street park and the green belt of the Yuebao Road Section, thereby embodying the harmonious concept of integration of man and nature.

The HYBAS process is a composite process involving biofilm and activated sludge. It has the advantages of an activated sludge process and a fluidized bed biofilm (MBBRTM) process. The two are organically combined in the same pool, and it is equipped with a double-layer advection-type secondary sedimentation. The pool not only operates stably and ensures that the effluent water quality reaches the standard of Class A or above, but can also effectively save the land used for project construction; thus, the actual land occupation per ton of water of the project is only 0.23 $m^2$/ton, which is much smaller than that of the same scale sewage treatment plant in the "Urban sewage treatment project", whose land occupation index according to the Project Construction Standard was 0.75 $m^2$/ton/day. The advanced sewage treatment unit uses a D-type filter and the phosphorus removal effect is improved by adding PAC.

2.    Kunming Ninth Sewage Treatment Plant—MBR Process

The Kunming Ninth Sewage Treatment Plant is one of the key projects in the "Twelfth Five-Year Plan" for Dianchi Lake treatment. It was officially put into operation in the beginning of 2014. Its operation further improved the overall sewage treatment capacity of Kunming City which has increased to 1.91 million tons per day, effectively reducing the pollution load discharged into Dianchi Lake. The total investment for the Kunming Ninth Sewage Treatment Plant is CNY 650 million, of which the total construction investment is

close to CNY 450 million and the construction investment per ton of water is CNY 4500, which can be reduced to about CNY 4000 after process optimization. The sewage treatment plant covers an area of 2.8 hectares, with a treatment capacity of 100,000 tons per day, a service area of 22.85 square kilometers, and a population of nearly 300,000.

The whole underground double-layer design is adopted for this plant. The sewage treatment facilities and maintenance units are located on the second floor and the first floor of the basement, respectively. Only one office building is on the ground level, and the other spaces are built in the surrounding open landscape parks, thereby achieving environment harmony and unity and improving the comprehensive quality of the city. Kunming Ninth sewage treatment plant adopts the membrane bioreactor (MBR) treatment process. Sewage is treated successively through screens, grit chambers, fine screens, biochemical tanks, and membrane tanks. Finally, it is lifted using the produced water pump and discharged after disinfection.

The MBR process has the advantages of high treatment efficiency, good effluent quality, compact equipment, and a small footprint. It is very suitable for use in underground sewage treatment plants, making the main indicators of effluent water quality of the plant better than the "Pollutant Discharge of Urban Sewage Treatment Plants" (GB18918-2002) Class A standard, similar to Class II water standard. The 40,000 tons/day reclaimed water produced by the Ninth Sewage Treatment Plant can be used for surrounding landscaping and miscellaneous municipal purposes, effectively improving the efficiency of water resource recycling.

3.    Beijing Daoxianghu Reclaimed Water Plant—Segmented Water Inlet $A^2O$ Process

The Beijing Daoxianghu Reclaimed Water Plant is located in the northwest of Haidian District, Beijing. Its construction officially started in August 2013 and was put into operation in 2016. The water plant is based on the reconstruction of the original above-ground sewage treatment plant. It covers an area reduced from the original 17 hectares to the current 4.47 hectares, a reduction of nearly three quarters. The treatment capacity is 160,000 tons per day, the service area is about 34.5 $km^2$, the investment in the first phase of the project is about CNY 480 million, and the treatment capacity is 80,000 tons/day. This plant is currently the largest underground sewage treatment plant and the largest reclaimed water plant in the northern region of China.

The Daoxianghu reclaimed water plant adopts the $A^2O$ denitrification and phosphorus removal technology in the influent. The problem of insufficient sources makes the effluent meet the quasi-IV surface water standard. On one hand, the treated reclaimed water is directly used for water landscaping of the ground garden of the factory area, and on the other hand, it is used for water replenishment of rivers and lakes and miscellaneous urban water bodies.

At the same time, the water plant adopted technologies such as rectangular sedimentation tanks entering and exiting each other, multi-stage biological deodorization technology, water source heat pump heat recovery technology [6], precise aeration, precise dosing and natural lighting, etc., to achieve a low carbon run.

The water reclamation plant makes full use of the vertical space and is divided into landscape park, covering soil layer, maintenance operation layer and sewage treatment layer from top to bottom. The negative basement floor is the maintenance operation layer, and the top is covered with soil for comprehensive utilization as urban green space resources; the lower structure is the main body of sewage treatment, wherein the sewage treatment facilities are set up. This plant played a very good role in improving the surrounding ecological environment, residents' public service facilities and cultural development. At the same time, it has help carry out high-end technology research and development mission of the BEWG [7–11].

The ground space of the Daoxiang Lake Reclaimed Water Plant is positioned to create a low-carbon demonstration park with distinctive characteristics, including an ecological park, a vitality park, a cultural park and a high-end technology research and development base, making it a green engine for regional development and thus improving the ecological

environment. The surrounding land has appreciated in value and has become a public service center and an activity center for the citizens and factory employees, focusing on displaying traditional Chinese culture and local culture, and carrying forth the high-end technology research and development mission of BEWG. It is understood that the Daoxianghu Reclaimed Water Plant will rely on the comprehensive cooperation between the Beijing Enterprises Water Group, Tsinghua University, Renmin University of China, Beijing University of Technology and other top enterprises and colleges to build it into a domestic first-class "recycled aquatic product research base" [11–13].

*2.2. Semi-Underground*

1. Beijing Xiaojiahe reclaimed water plant—A$^2$O process

The Beijing Xiaojiahe reclaimed water plant carried out an in situ upgrade on the basis of the original Xiaojiahe sewage treatment plant. Without adding an inch of land, the original 20,000 tons/day treatment capacity was upgraded to 80,000 tons/day after the transformation. The effluent water quality has been improved from the original Class A to the Beijing Landmark B standard, the main indicators of which meet the surface water category IV water standard, and the treated clean water can be directly injected into the Qinghe River, which can provide high-quality and safe recycled water sources [14–16].

The Xiaojiahe reclaimed water plant is a BOT project built by the Sound Group, covering an area of 2 hectares. The upgrading project started in August 2013 and is currently in progress. The water plant adopts a semi-underground construction model, wherein all sewage treatment facilities are buried underground, and productive buildings, offices, and living areas are built on the ground to minimize the harmful impact on the environment. After completion, the greening rate of the plant area becomes greater than 30%. Compared with the area of the original sewage treatment pool which only accounted for 15% of the plant area, the area of the sewage treatment pool of the reconstructed underground sewage treatment plant has increased by nearly four times [17].

After the transformation, the sewage treatment adopts the "A$^2$O-energy-saving MBR" process. The pulse aerator used is independently developed by the Sound Group, which can mechanically produce pulse bubbles that can make the fiber membrane shake intermittently to prevent the membrane pores from clogging, can greatly reduce the required blast volume, and save the electricity required for tons of water treatment by 0.15 kWh, making the energy consumption of the process less than 1/3 of the membrane aeration energy consumption compared with the traditional MBR process, and thus save tens of thousands of kWh of electricity per day. The effluent adopts an ozone-ultraviolet combined oxidation and disinfection method, and is equipped with odor collection and biological deodorization devices, which will hardly cause pollution during the production process and hamper the life of residents in the surrounding areas [18–21].

2. Nanpian sewage treatment plant in Wenzhou city—BAF Process

The Nanpian Sewage Treatment Plant has a total land area of 8.72 hectares and a total sewage treatment scale of 80,000 tons/day. The first phase of the project covers an area of 4.12 hectares, with an estimated total investment of CNY 249 million and a treatment scale of 40,000 tons/day. The construction started in early 2013 and it started operating in April 2015. The project construction lasted 2 years.

The Nanping treatment plant is the first "high standard, full deodorization, combined, full coverage" garden sewage treatment plant in the Zhejiang Province. It follows the semi-underground design, wherein the sewage treatment facilities are all buried underground, and the repair and maintenance buildings, office and living areas are unified on the ground. The factory area is covered with garden-style greening, except for the roads. A green "roof" is added on the roof of the main factory building and the comprehensive building, and local Wenzhou tree species, sweet-scented osmanthus, magnolia and other fragrant plants are planted to make the factory area fragrant [22]. All these corporations have allowed people

to have a new impression of the treatment plant as a green park full of vitality, completely overturning their previous views of sewage treatment plants as being dirty and smelly.

For sewage treatment, this plant adopts the French Veolia biological aerated filter process (BAF) [23–25]. The main process flow consists of a Mutiflo high-efficiency sedimentation tank plus Biostyr aerated biological filter and Actiflo sand-added high-efficiency sedimentation tank, which have the advantages of good treatment effect and stable operation [26–30]. The modular combination of each processing structure effectively saves the land used for equipment, making the construction area of the main factory building in the first phase (including some public auxiliary facilities in the second phase) only cover an area of more than 7000 square meters, and the rest is used for landscape greening design. The water plant has a high degree of automation and can realize fully automatic operation; the main plant adopts a fully enclosed deodorization design, and the odor is collected through pipelines and sent to the biological deodorization system for treatment before being discharged, without affecting the surrounding environment [31–33].

### 2.3. Fully Enclosed above-Ground

The initial phase of the sewage treatment plant in the east district of Guangzhou economic and technological development zone has an investment of about CNY 70 million [34–36]. The sewage treatment capacity is 25,000 tons/day [37]. The construction investment per ton of water is close to CNY 2800/ton, and the sewage treatment cost is about CNY 1.14/ton [38–40]. The service scope of this plant is the eastern district of Guangzhou economic and technological development zone, with a total service area of 7 square kilometers [40–42]. The eastern district sewage treatment plant occupies a relatively small area, and the first phase covers an area of about 1.6 hectares [43]. The water plant adopts the CASS process and completes the denitrification and phosphorus removal process through four stages of water inflow agitation-aeration-sedimentation-skimming in a period of 6 h. The automatic control system adopts the most advanced, remote three-level control and realizes online monitoring of turbidity, PH, dissolved oxygen, liquid level, and flow [44]. This water plant is the first sewage treatment plant in the country to be covered with steel structure, which not only minimizes the noise and odor impacts on the surrounding environment but also makes the appearance of the sewage plant feel like a modern factory [45].

### 2.4. Open on the Ground

The Beijing Xiaohongmen sewage treatment plant is the second largest sewage treatment plant in Beijing, with a planned drainage area of 223.5 km$^2$ and a service population of about 1.925 million [46,47]. The water plant adopts the A$^2$O process, with a treatment capacity of 600,000 tons per day, a total investment of about CNY 1.2 billion, an investment cost of about CNY 2000 per a ton of water, and a sewage treatment cost of about CNY 1.12 per ton of sewage [47–49]. The water plant covers a total area of 47 hectares. The main construction contents of the plant area include plant civil construction, process equipment, process pipeline installation, electrical and automatic control system installation, lighting, lightning protection grounding, heating, ventilation, plant road construction, greening, etc. The sewage passes through the grille, lifting pump room, aerated grit chamber, primary sedimentation tank, A$^2$O tank (anaerobic to anoxic to aerobic tank volume ratio is 1:3:6.7), and the secondary sedimentation tank, and then is disinfected and discharged outside the plant; the sewage discharge standard is "water pollutant discharge standard for urban sewage treatment plants" [49,50].

The Beijing Xiaohongmen sewage treatment plant has effectively improved the water quality of the Liangshui River and the surrounding ecological environment. In carrying out sewage treatment, this plant has also strengthened the construction of reclaimed water. The reclaimed water treatment capacity in the plant is 1500 cubic meters per day, which is used in the plant area for greening, toilet flushing, landscape and equipment flushing.

Since April 2006, it has also supplied 300,000 tons of recycled water per day to the Daxing Liangfeng irrigation canal as irrigation water for farmland in the area [51].

## 3. Methodology and Developmental Processes

The total occupied land is relatively small due to the underground design of the sewage treatment plant, so the requirements for the size and performance of equipment needed for the main processes of sewage treatment are higher. Some traditional sewage treatment processes used in above-ground sewage treatment plants are generally not used as the main body of underground sewage treatment plants or used alone due to factors such as large areas, general denitrification and phosphorus removal efficiency, and large surplus sludge output. Approaches with a small footprint, large volume load, high treatment efficiency, and less residual sludge are widely used in the secondary or advanced treatment process of underground sewage treatment plants [52]. The following is an inventory of the technical process routes commonly used in underground sewage treatment plants in China.

### 3.1. MBR Process

Membrane bioreactor (MBR) is a new type of water treatment technology that combines an activated sludge process and a membrane separation technology. The MBR process has the advantages of high-quality and stable effluent (Grade A, and can reach the standard of surface IV water), less residual sludge production (1/2 of the traditional process), small footprint, high efficiency of nitrogen and phosphorus removal, automatic control and so on. Hence, it is one of the most advanced sewage treatment and recycling processes in the world.

The MBR process is especially suitable for the sewage treatment processes of underground sewage treatment plants with limited floor space and high automation requirements and has obvious advantages over other treatment processes based on its characteristics mentioned above. At present, many of the fully underground sewage treatment plants under construction already built in China have adopted the MBR process (Table 1). This process is highly intensive, has little impact on the surrounding environment, and the treated effluent can be directly used to reload water in landscape river [53]. In addition, the MBR membrane filtration process is used to replace the traditional secondary sedimentation tank for advanced treatment, which can increase the sludge concentration in the biochemical tank while saving land, thereby improving the treatment efficiency of the system [54].

**Table 1.** Summary of the MBR process in underground sewage treatment plants.

| No | USTP Name | Scale (10 k Tons/Day) | Main Body | Water Quality |
|----|-----------|----------------------|-----------|---------------|
| 1 | Kunming Tenth Sewage Treatment Plant [55] | 15 | MBR | Higher than A |
| 2 | Shijiazhuang Zhengding New Area Buried Recycled Water Plant [56] | 10 | MBR biochemistry + MBR advanced treatment | Better than Class I A; Partial Class IV |
| 3 | Kunming Ninth Sewage Treatment Plant [55] | 10 | MBR | Higher than A |
| 4 | Suzhou Zhangjiagang Jingang Area Sewage Treatment Plant [57] | 5, Expect 2.5 | Improvement $A^2O$ + MBR | Higher than A |
| 5 | Guangzhou Jingxi Underground Water Purification Plant [58] | 10 | MBR | Level A |
| 6 | Phase I of Tiantanghe Sewage Treatment Plant in Daxing District, Beijing | 8, Expect 4 | $A^2O$ + Multi-segment AO + MBR | Level A |

**Table 1.** *Cont.*

| No | USTP Name | Scale (10 k Tons/Day) | Main Body | Water Quality |
|---|---|---|---|---|
| 7 | Hefei Binhu New District Tangxi River Reclaimed Water Plant [59] | 3 | MBR | Level A |
| 8 | Yantai Taoziwan Sewage Treatment Plant Phase II Project | 15 | MBR | L A and 100,000 tons/day recycled water |
| 9 | Taiyuan Jinyang Underground Sewage Treatment Plant | 48, Expect 32 | Improvement A$^2$O (2,010,000 tons/day) + MBR (1,210,000 tons/day) | Level A |
| 10 | Xiaojiahe Reclaimed Water Plant Project | 8 | A$^2$O + energy saving MBR | Class A to IV water standards |

Based on industry experts' views, the MBR core technology of underground sewage treatment plants are relatively mature at present, and the area per ton of water is 0.26 m$^2$, and the investment per ton of water is about CNY 4000–5000. For areas with developed economy, scarce land resources and high environmental requirements, underground sewage treatment plants with MBR as the main treatment process will become the new development trend.

*3.2. A$^2$O, Improved A$^2$O and Its Combination Process*

Anaerobic–anoxic–oxic (A$^2$O) is a commonly used sewage treatment process, which can be used in secondary and tertiary sewage treatment and reclaimed water reuse, and has a good denitrification phosphorus removal effect. In order to further improve the processing capacity of the A$^2$O process, an anoxic pool can be added before the anaerobic pool at the front end of the A$^2$O process to enhance the effect of denitrification. In this article, these upgraded A$^2$O processes are collectively referred to as the improved A$^2$O process. The improved A$^2$O process has the advantages of high nitrogen and phosphorus removal efficiency, effective inhibition of filamentous bacteria expansion, and low operating costs. In order to obtain higher and more stable effluent quality, underground sewage treatment plants use the improved A$^2$O as the main process in combination with other process equipment during design, such as MBR, MBBR, deep bed filter, and fiber rotary disc filter. Please refer to Table 2.

Among the various A$^2$O and its combined processes listed in Table 2, the A$^2$O + biofilm combined pool (HYBAS) process adopted by the Shenzhen Buji underground sewage treatment plant and the segmented influent A$^2$O used by the Beijing Daoxianghu reclaimed water plant have distinctive features. Of the two, the HYBAS process has the advantages of the activated sludge process and the fluidized bed biofilm (MBBRTM) process. Combining the two organic approaches in the same tank can significantly increase the sludge concentration and volume load, thereby improving the organic matter removal of the system and denitrification and phosphorus removal efficiency; the segmented influent A$^2$O process mainly solves the problem that the traditional A$^2$O process finds difficult, which is simultaneous denitrification and stable removal of phosphorus. At present, this process has been applied as the main process of the Beijing Daoxianghu underground sewage treatment plant.

**Table 2.** Summary of A$^2$O and its improved processes in underground sewage treatment plants.

| No | USTP (Name) | Scale (10,000 Tons/Day) | Main Body | Water Quality |
|----|-------------|-------------------------|-----------|---------------|
| 1 | Guiyang Qingshan Reclaimed Water Plant | 5 | Improved A$^2$/O (JHB) | COD, ammonia nitrogen, TP, surface water category IV, and other indicators are level A standard |
| 2 | Guiyang Madi River Reclaimed Water Plant | 3 | Improved A$^2$/O | COD, ammonia nitrogen, TP, surface water category IV, and other indicators are level A standard |
| 3 | Shenzhen Buji Sewage Treatment Plant [60] | 20 | A$^2$O+ biofilm co-pool (HYBAS) | Level A standard |
| 4 | Kunming Anning Second Sewage Plant | 6 | A$^2$O | Level A standard |
| 5 | Sewage Treatment Plant of Taiping Town, Anning City, Kunming | 2.5 | A$^2$O | Level A standard |
| 6 | Qingdao High-tech Zone Sewage Treatment Plant [61] | 18, Phase one 9 | A$^2$O + MBBR | Level A; |
| 7 | Hefei Shishilihe Sewage Treatment Plant Phase II | 10, Phase one 5 | A$^2$O | Better than Class I A; Partial Class IV |
| 8 | Yantai Guxian Sewage Treatment Plant Phase II | 6 | Inverted A$^2$O | Level A |
| 9 | Suzhou Zhangjiagang Jingang Area Sewage Treatment Plant | 5, Phase one 2.5 | Improved A$^2$O + MBR | Better than level A |
| 10 | Phase I of Tiantanghe Sewage Treatment Plant in Daxing District, Beijing | 8, Phase one 2, Phase two 6 | A$^2$O (Phase 1) + multi-segment AO (Phase 2) + MBR filter | Level I B, Part I Level A |
| 11 | Beijing Daoxianghu Reclaimed Water Plant | 16, Phase one 8 | Segmented water intake A$^2$O | Surface level IV |
| 12 | Taiyuan Jinyang Underground Sewage Treatment Plant | 48, Phase one 32 | Improvement A$^2$O + MBR | Level A |
| 13 | Xiaojiahe Reclaimed Water Plant Project | 8 | A$^2$O + energy saving MBR | Class A to IV water standards |

*3.3. Other Sewage Treatment Processes*

1.    Moving bed biofilm reactor (MBBR) process

The MBBR is a process in which a certain amount of suspended carriers are added to the reactor for microbial implantation to form a biofilm to facilitate biological nitrogen and phosphorus removal and organic matter removal. The MBBR process has the advantages of both the traditional fluidized bed and the biological contact oxidation methods. The sludge yield is low, thus removing the need to set up sludge return equipment, or packing supports, and thereby overcomes the limitations of the traditional activated sludge method and the fixed biofilm method. This process is widely used because of its low investment and convenient operation and management. The underground sewage treatment plant in Qingdao High-tech Zone is one of the most representative plants employing MBBR engineering processes. Although the cost per ton of water of the underground design is slightly higher than that of the above-ground design of the same scale, the MBBR process takes up less space and the operation and management costs are lower. Thus, the comprehensive operating cost of the entire underground sewage treatment plant does not have obvious disadvantages compared with the above-ground design.

2.    Multi-stage AO process

The multi-stage AO process (Biolak process) requires setting up multiple anoxic and aerobic areas in the same structure so that the sewage can realize the multi-stage anoxic and aerobic biochemical reaction processes, and so that the pre-anaerobic reaction area reaches a relatively high level. In the multi-stage AO process, water first enters the anaerobic zone, then flows to the anoxic zone, facultative oxygen zone, and aerobic zone of the biological pool so that the carbon source in the sewage can be selectively supplied to different functional zones preferentially used for anaerobic phosphorus release and denitrification, which can effectively save carbon sources. Furthermore, it can also increase the sludge concentration and reduce the volume of the biological pool. The multi-stage AO series connection can cancel the need for internal reflux equipment, and the digestion liquid of the upper stage can completely enter the anoxic zone of the lower stage for denitrification to strengthen the denitrification effect and efficiency. The multi-stage AO process with multi-stage water inflow is one of the improved processes of $A^2O$, which can distribute carbon sources more reasonably and reduce the investment and operating costs. It has been applied in the underground sewage treatment plant in Guang'an City, Sichuan Province, China.

## 4. The Development of Underground Sewage Treatment Plants

In recent years, underground sewage treatment plants have gained more and more attention in urban sewage treatment development in China and abroad because of their advantages over the traditional above-ground sewage treatment plants, which include less land occupation, less noise and odor pollution, and better landscape. The history of construction and application of underground sewage treatment plants is more than 80 years [62]. Finland built the world's first underground sewage treatment plant in 1932; most of Sweden's sewage treatment plants are underground, leading the world both in terms of quantity and quality. In addition, countries such as Japan, South Korea, the Netherlands, the United Kingdom, Norway, France, Switzerland, Monaco, and the Czech Republic have also built underground sewage treatment plants, which have effectively resolved the contradiction between sewage treatment, environmental pollution, and urban land use, and achieved great results in terms of economic and social benefits. In China, there was comparatively a delayed construction of underground sewage treatment plants. Hong Kong operated the first underground sewage treatment plant with a treatment capacity of 12,000 tons per day in 1995. Large and medium-sized cities in some provinces of China have gradually begun to build underground sewage treatment plants, the smallest of which is the Guilin City Leping sewage treatment plant (10,000 tons/day) and the largest operating scale is that of the Shenzhen Buji sewage treatment plant (200,000 tons/day); the scale of Beijing Huaifang underground sewage treatment plant under construction has reached 600,000 tons/day.

*Status Quo of Development and Application*

Cities in developed countries such as the United States, Japan, and South Korea attach great importance to the use of underground space. Underground treatment plants are preferred due to two primary reasons: to prevent the impact of extremely cold weather on the sewage treatment effect, such as in Sweden, Finland, Norway, and other Nordic countries; and to save urban land resources, such as in Singapore, Japan, South Korea, the Netherlands, and other countries. Over the years, these countries have considerably developed and utilized the underground spaces, and urban underground drainage and sewage treatment systems therein have also made great progress. A considerable number of underground sewage treatment plant projects have been built and obtained good environmental, economic, and social benefits. Table 3 summarizes some of the low-water treatment plants operating in developed countries.

**Table 3.** Summary of some underground sewage treatment plants.

| Nation | Name | Scale (10,000 Tons/Day) | Design Form |
|---|---|---|---|
| Japan | Chikumagawa Underground Sewage Treatment Plant in Nagano Prefecture | | |
| | Underground sewage treatment plant in Hayama Town, Kanagawa Prefecture [63] | 2.47 | Mountain tunnel |
| | Kashima Town Sewage Treatment Plant, Shimane Prefecture | | Mountain tunnel |
| | Gifu City Sewage Treatment Plant | 17.52 | Semi underground |
| | Sendai Kamo Sewage Treatment Plant | | All underground |
| | Tokyo City Ukima Recycled Water Company | 10 | All underground |
| South Korea | Busan Underground Sewage Treatment Plant | 10 | All underground |
| | Daegu Jisan Sewage Treatment Plant | 100,000 PE | All underground |
| | Yongin Underground Sewage Treatment Plant | 11 | |
| | Incheon Wastewater Treatment Plant | 12.5 | Semi underground |
| Sweden | Henriksdal Sewage Treatment Plant | 25 | mountain tunnel |
| | Bromma Wastewater Treatment Plant | 14 | full underground |
| | Loudden Wastewater Treatment Plant | 1 | full underground |
| Finland | Wiggin McKee Center Wastewater Treatment Plant | 32.8 | full underground |
| Norway | Veas Wastewater Treatment Plant | 25 | full underground |
| Netherlands | Dokhaven sewage treatment plant | 34 | full underground |
| England | Xinqi Underground Sewage Treatment Plant | 21.6 | full underground |
| France | Marseille sewage plant | | full underground |
| | Toulon Underground Sewage Treatment Plant | 3.4 | Mountain tunnel |
| | Antipu Underground Sewage Treatment Plant | | full underground |
| Switzerland | Geneva underground sewage treatment plant | | semi underground |
| Italy | Media Pusteria Underground Sewage Treatment Plant | | |
| Monaco | Underground sewage treatment plant in the Kingdom of Monaco | | |
| Czechoslovakia | The First Underground Sewage Treatment Plant in the Czech Republic | | Tunnel |

Finland is the first country in the world to build underground sewage treatment plants [2]. It began the construction of underground sewage treatment plants in 1932, but due to technical limitations, it failed to develop further. Until 1991, the underground space development technology was relatively nascent, and it built on the premise of environmental protection, land resources, and economic benefits. Finland decided to build a large-scale underground sewage treatment plant in the center of the capital Helsinki—Viking. The construction of the McKee Center sewage treatment plant, now used to treat municipal domestic sewage and reduce pollution in the Gulf of Finland, was completed in 1994, with a design scale of 328,000 tons per day and a total project cost of USD 215 million of which the cost of the underground sewage treatment plant itself was USD 198 million. The development of the treatment plant has accelerated the establishment of underground sewage treatment plants in Finland, effectively realizing environmental protection, land resource conservation, and economic benefits, and thus is of great significance to improving the living environment of Helsinki and preventing environmental pollution.

Sweden also developed an underground sewage treatment plant more than 70 years ago. In 1942, Stockholm, the capital of Sweden, took advantage of the local superior geological conditions and advanced rock excavation technology to build the world's first modern sewage treatment plant. For underground sewage treatment plants, the ground of the whole plant area is arranged as a park. The entrance of the underground sewage treatment plant adopts an ingenious architectural art, which greatly beautifies the city's appearance and at the same time increases the city's green area [4]. Most of the sewage treatment plants in Sweden adopt the underground construction model. Although most of them are built in densely populated areas, the underground construction model ensures that the lives of the residents in the surrounding areas are basically not affected in any way.

Korea is also at the forefront of the world in the application of underground sewage treatment plants. In recent years, in order to save land, most of the newly built underground sewage treatment plants in South Korea have adopted the underground construction model, which is said to account for about 50% of all sewage treatment plants in South Korea. One of the more representative cases is the upgrading and renovation project of the Suyeong above-ground sewage treatment plant in the center of Busan. With the rapid development of social economy, Seoul, South Korea is facing various problems such as higher requirements for effluent water quality, shortage of urban land, and increasing requirements for the living environment of people around the water plant. It is imperative to change the original Suyeong sewage treatment plant into an underground type. It is also listed as one of the important contents of urban sewage treatment infrastructure construction in Busan in the next 20 years. The upgraded underground sewage treatment plant adopts a three-stage treatment scheme of pretreatment + biochemical treatment + membrane filtration, with a designed treatment scale of 100,000 tons per day.

The Dokhaven sewage treatment plant in the Netherlands is located in the center of Rotterdam, the second largest city in the Netherlands. It is the only underground sewage treatment plant in the Netherlands and only the central control room is built on the ground. This sewage treatment plant has been in operation for 28 years and was built in 1987. The sewage treatment adopts the AB process and the sludge generated in the mainstream section is transported to the Sluisjesdijk sludge treatment plant 600 m away for anaerobic digestion. The reason for choosing the AB process is that its implementation occupies a small area, and its compactness is just suitable for the lack of planned land for construction, and the effluent standard at that time had no restrictions on nitrogen and phosphorus. The underground construction model was chosen because the land where the water plant is located is multi-purpose and has little impact on the surrounding environment.

It can be seen that rich construction experience, leading technologies, and advanced landscape design concepts have enabled some underground sewage treatment plants in the above-mentioned countries to achieve economic and social benefits. They have also achieved huge environmental benefits, and thus can serve as guidelines for building underground sewage treatment plants in China.

## 5. China's Development and Application Status

### 5.1. Background of Underground Sewage Treatment Plants in China

1. The cost of renovation and relocation of above-ground sewage treatment plants is huge

The noise, odor and other pollution generated during the operation of the above-ground sewage treatment plants will adversely affect the physical and mental health of the residents in the surrounding areas. With the continuous development of China's urban economy and the need for improvement in people's requirements for a better quality of living environment, the sewage treatment plants previously built in the city center now faces several problems and thus require to be renovated and relocated, the cost of which is huge. The underground sewage treatment plant can be built in the center of the city and because of its closed design, it can prevent noise, odor and other pollution, all of which make them suitable for the sewage treatment needs of future urban development.

2.　　High water quality and quantity requirements for river replenishment

With the development of the economy and the continuous increase in the total amount of urban sewage discharge, the water quality of the water environment is deteriorating day by day. Urban rivers have become polluted and thus no longer can play their due role in beautifying the environment. The existing above-ground sewage treatment plants have low sewage treatment efficiency and the effluent water quality is also low. Underground sewage treatment plants generally adopt more advanced treatment processes and equipment, and the effluent water quality is above grade A or higher, which can be used to replenish water in urban rivers and improve the quality of the water environment.

3.　　Rapid growth of urban land prices

The construction of above-ground sewage treatment plants will occupy a large amount of urban land resources. With the continuous increase in urban land prices, the construction cost of above-ground sewage treatment plants will be significantly increased. Furthermore, the environmental pollution of the above-ground sewage treatment plants will also cause the depreciation of surrounding land resources. In addition, isolation belts will have to be set up between the factory area and surrounding residential areas and commercial areas, which will waste land resources and affect the image of the city.

In this context, people gradually realize that an underground sewage treatment plant that takes up less space, saves land resources, has less environmental pollution, and can be coordinated with the surrounding environment to meet the needs of urban development and water pollution prevention in China. Underground sewage treatment plants have achieved rapid development in recent years.

*5.2. Development Status*

As compared to the rest of the world, the underground sewage treatment plant strategies developed rather late in China. In addition, China's water treatment industry as a whole has a relatively limited understanding of the advantages and development prospects of underground sewage, and hence this industry obtains very little capital investment. Since the 1990s, due to the increasing demand for intensive land use and environmental landscape in cities, the construction of underground sewage treatment plants has been promoted to a certain extent. Underground sewage treatment plant projects have a clear upward trend in China, providing a new model for the construction of urban sewage treatment facilities and recycling of water resources.

Although the total number of underground sewage treatment plants in China has increased a lot compared with the past, the proportion of underground sewage treatment plants is still very small in comparison to the current above-ground sewage treatment plants in the country which is over 3800. At present, the choice of building an underground sewage treatment plant is mainly based on the consideration of planning, land, environment, and other constraints. Although the number of underground sewage treatment plants accounted for a relatively low proportion (~3%) of China's sewage treatment at the end of the "Twelfth Five-Year Plan" period, the growth trend was obvious throughout this period. The amount of treated water and total investment has increased by about 10.4% and 20.8% in sewage treatment plants, respectively.

Underground sewage treatment plant projects have been launched in Beijing, Shanghai, Guangzhou, Kunming, Hefei, Qingdao, Suzhou, Guilin, Xiamen, Shijiazhuang, Wenzhou, Taiyuan and other cities so far, and nearly 40 underground sewage treatment plants have been put into operation or are under construction; the total processing capacity of the plants is near to 4.9 million tons/day. There are 22 underground sewage treatment plants in operation, with a total treatment capacity of nearly 1.9 million tons, as shown in Table 4. There are 17 underground sewage treatment plants under construction, with a total treatment capacity of more than 3 million tons. The treatment scale of the plants displays an increasing trend, as shown in Table 5. Most of these underground sewage treatment plants are designed and constructed by the four major sewage treatment compa-

nies, including China Water Environment Group, Beijing Enterprises Water, Origin Water, and Sound Group. According to statistics, these four major enterprises have more than 20 underground sewage treatment plant projects under construction or in operation.

**Table 4.** Overview of the 22 underground sewage treatment plants in operation in China.

| No | Name | Scale (10,000 Tons/Day) | Invest (100 Million Yuan) | Floor Area (Hectare) | Operation Time |
|---|---|---|---|---|---|
| 1 | Guiyang Qingshan Reclaimed Water Plant | 5 | 3.20 | 2.11 | 2014 |
| 2 | Guiyang Madi River Reclaimed Water Plant | 3 | 1.65 | 1.63 | 2014.12 |
| 3 | Shenzhen Buji Sewage Treatment Plant | 20 | 5.90 | 4.60 | 2011.8 |
| 4 | Kunming Anning Second Sewage Plant | 6 | | | |
| 5 | Sewage Treatment Plant of Taiping Town, Anning City, Kunming | 2.5 | | | |
| 6 | Qingdao High-tech Zone Sewage Treatment Plant | 18 | 5.80 | 7.34 | 2014.7 |
| 7 | Kunming Tenth Sewage Treatment Plant | 15 | 7.00 | 3.94 | 2013.7 |
| 8 | Shijiazhuang Zhengding New Area Underground Reclaimed Water Plant | 10 | | 2.00 | 2015.6 |
| 9 | Hefei Shishilihe Sewage Treatment Plant Phase II | 10 | | 3.05 | 2014.1 |
| 10 | Kunming Ninth Sewage Treatment Plant | 10 | 6.50 | 2.80 | 2014.9 |
| 11 | Yantai Guxian Sewage Treatment Plant Phase II | 6 | 1.60 | | 2013.8 |
| 12 | Suzhou Zhangjiagang Jingang Area Sewage Treatment Plant | 5 | 2.89 | | 2012.12 |
| 13 | Guangzhou Jingxi Underground Water Purification Plant | 10 | 5.80 | 1.87 | 2010.9 |
| 14 | Guilin Pingle Sewage Treatment Plant | 1 | 0.50 | | 2010.6 |
| 15 | Beijing Daxing Tiantanghe Sewage Treatment Plant | 8 | 1.31 | 5.04 | 2009.2 |
| 16 | Hefei Binhu New District Tangxi River Reclaimed Water Plant | 3 | 1.47 | | 2008 |
| 17 | Taipei Neihu Sewage Treatment Plant | 15 | | | 2002.11 |
| 18 | Hong Kong Stanley Sewage Treatment Plant | 1.2 | | | 1995 |
| 19 | Beijing Daoxianghu Reclaimed Water Plant | 16 | 4.8 0 | 4.47 | 2015.7 |
| 20 | Kunming Eleventh Sewage Treatment Plant | 6 | 3.80 | 4.07 | 2015 |
| 21 | Kunming Twelfth Sewage Treatment Plant | 10 | | | 2015 |
| 22 | Nanpian Sewage Treatment Plant in Wenzhou City | 8 | 2.49 | 4.12 | 2015.4 |

**Table 5.** Overview of the 17 underground sewage treatment plants under construction in China.

| No | Name | Scale (10,000 Tons/Day) | Form of Investment and Total (100 Million Yuan) | Floor Area (Hectare) | Estimated Commissioning Time |
|---|---|---|---|---|---|
| 1 | Beijing Tongzhou Bishui Reclaimed Water Plant | 18 | | | 2016.6 |
| 2 | Shanghai Jiading Nanxiang Reclaimed Water Plant | 15 | PPP, BOT | | |
| 3 | Guiyang Sanqiao Reclaimed Water Plant | 4 | | | |
| 4 | Guiyang Guancheng River Reclaimed Water Plant | 6 | | | |
| 5 | Guang'an Reclaimed Water Plant | 5 | | | |
| 6 | Guang'an Kuige Reclaimed Water Plant | 2 | | | |
| 7 | Singapore Changi No. 2 NEWater Plant | 22.8 | DBOO | | 2016 |
| 8 | Malaysia Pantai Sewage Treatment Plant | 32 | 20 | 10.5 | |
| 9 | Yantai Taoziwan Sewage Treatment Plant Phase II | 15 | | 9.49 | |
| 10 | aiyuan Jinyang Underground Sewage Treatment Plant | 48 | 25 | 27.35 | 2015.12 |
| 11 | Xiaojiahe Reclaimed Water Plant | 8 | BOT | 2 | |
| 12 | Fuqing Second Sewage Treatment Plant | 5 | BOT | 2 | |
| 13 | Hefei Qingxi Water Purification Plant | 20 | BOT, 10.54 | 5.34 | |
| 14 | Xiangtan River East Second Sewage Treatment Plant | 15 | PPP, 4.8 | 4.54 | |
| 15 | Beijing Beiyuan Underground Sewage Treatment Plant | 5 | | | |
| 16 | Beijing Huaifang Reclaimed Water Plant | 60 | | 31 | 2016 |
| 17 | Urumqi Hexi Sewage Treatment Plant | 20 | 7 | | |

The construction of underground sewage treatment plants in China is booming and related technologies are also playing an increasingly important role in the field of international sewage treatment. Among them, the Pantai underground sewage treatment plant in Malaysia [64] and the Changi No. 2 NEWater plant in Singapore are notable representatives [65].

The Pantai underground sewage treatment plant is located in Kuala Lumpur. It adopts all the structures of an underground treatment model. The designed scale of sewage treatment is 320,000 tons per day, covering an area of 10.5 hectares. It is the first underground sewage treatment plant invested by an overseas Chinese company. The project adopts the improved $A^2/O$ denitrification and phosphorus removal process, and the $ClO^2$ disinfection scheme after using the low-load secondary sedimentation tank effluent as a standard. The sludge is digested and concentrated, then dehydrated and transported outside for disposal. This plant is majorly utilized as a biogas power generation plant, and the waste heat is used for refrigeration; the regenerated water is filtered and decolorized using ozone through the "ultrafiltration membrane" for recycling of ground landscape water. The functional division of the Pantai sewage treatment plant project is equipped to protect the environment and effectively utilize the upper space of the sewage plant; the plant's layout is reasonable and compact, which meets the requirements of production, life and consumption, and is coordinated with an upper leisure park. The production area is completely separated from the park leisure area. Management can provide nearly an area of 140,000 $m^2$ of leisure parks and green landscapes.

The Singapore Changi No. 2 NEWater Plant DBOO project is currently under construction, with a project scale of 228,000 tons/day, and it uses the "microfiltration + reverse osmosis" method. The dual-mode technology process treats the effluent from the secondary

sedimentation tank of Singapore's Changi Sewage Reuse Plant into NEWater, which is discharged after ultraviolet disinfection treatment, and the NEWater is transported to the Singapore NEWater pipeline network system.

## 6. Problems and Challenges

In China, the decision to build an underground sewage treatment plant is based on concerns related to environmentally friendly conditions, land conservation, and resource reuse [66]. However, the following problems cannot be ignored.

1.  The construction cost

The construction of an underground sewage treatment plant requires deep foundation pit excavation and a layered layout [67], which is difficult and costly. In general, the investment cost of an underground sewage treatment plant (CNY 4000–6000/ton) is 2–3 times that of an above-ground sewage treatment plant of the same scale (CNY 2000/ton) [68,69]. The underground sewage treatment plant occupies a limited area, so it is necessary to choose a process and a corresponding equipment with a compact structure, high treatment efficiency, which require higher requirements for operation and maintenance, thus relatively increasing the overall cost of the process equipment, operation, and maintenance. The underground plant needs to add deodorization and ventilation equipment, whose operating costs are also high. The power consumption of these two items may reach 30–50% of the entire underground plant [70–72].

2.  High risks and great potential safety hazards

The toxic and harmful gases generated during the sewage treatment process are treated in airtight conditions and disposed-off before being discharged into the atmosphere due to the fully enclosed underground design, but there is a risk of odorous gas leakage, which will affect the health of underground workers [73]. Studies have pointed out that hydrogen sulfide, ammonia, and volatile organic compounds, such as methyl mercaptan and methyl sulfide, produced in the sewage treatment process of underground plants will cause serious harm to human health. The noise pollution in the sewage treatment process will also have an impact on the health of workers [74]. The potential safety hazards of underground sewage treatment plants cannot be ignored. Water back flow and sewage leakage may occur, resulting in the hidden danger of the water plant being flooded in case of electricity cut-off or heavy rainfall [75,76]. It is suggested that at least three gates should be installed in the underground sewage treatment plant to ensure the prevention of waterlogging. Furthermore, the underground sewage treatment plant also has fire hazards due to its underground closed construction form. It is necessary to set up corresponding fire protection zones according to different sub-areas of the underground structures [77].

3.  Single ground landscape design

The above-ground landscape design of the underground sewage treatment plant can increase the value of the above-ground space and its surrounding land, thereby offsetting the relatively high construction, operation, and maintenance costs generated by the underground construction, driving social and economic development around the water plant, and promoting ecological environment, environmental harmony and social development. Although the above-ground landscape design of underground sewage treatment plants in China has achieved certain developments, there are still problems, such as insufficient advanced concepts, relatively single models, and insufficient functions [78]. There is still a lot of room for improvement in the above-ground landscape design of underground sewage treatment plants in China. For example, diversification and functional design can be carried out around the carrier of underground sewage treatment plants and multi-functional places can be established [79].

4.  Lack of normative document guidance

Standardization is an essential factor to ensure the smooth development of underground sewage treatment plants, and the lack of corresponding standards and technical

specifications is also a problem faced by underground sewage treatment plants. It is known that in the United States, Japan, and the European Union, there are corresponding norms and standards for underground sewage treatment plants [32]. In China, the construction and operation, and maintenance of underground sewage treatment plants can only rely on traditional experience or one has to directly "move" the above-ground sewage treatment plants to deep underground foundation pits. There is no national technical standard to uniformly standardize the underground sewage treatment plants. However, the existing "urban sewage treatment project construction standards" and "urban sewage treatment plant pollutant discharge standards" can meet the construction, operation, and pollution index discharge standards of above-ground urban sewage treatment plants. The requirements of relevant regulations do not meet the specification requirements of engineering design, construction, equipment installation, operation and maintenance of underground sewage treatment plants [33].

5. Insufficient government subsidy policy and capital investment

Currently, the Chinese government has not issued a document to provide specific subsidies for the construction of underground sewage treatment plants. We only found that there are relevant subsidy policies in some places, such as repurchase at high prices for landscape water. For example, the Tangxi River Reclaimed Water Plant in Hefei City, Anhui Province uses membrane technology, consumes a lot of electricity, and the operating cost per ton of sewage treated is more than CNY 1. Therefore, the operation of the underground sewage treatment plant mainly relies on subsidies from the local government. Reclaimed water is purchased at a price of more than CNY 1 and is used as landscape water in wetland parks to ensure that the income and expenditure of the underground sewage treatment plant can basically be balanced and its long-term operation can be maintained [9]. Regarding capital investment, the investment in the construction of underground sewage treatment plants has increased significantly during the "Twelfth Five-Year Plan" period in China; however, there is still a significant gap compared with the investment in above-ground sewage treatment plants, as underground plants account for only 10% of the total investment. As China's urban land resources become increasingly rare and people's requirements for the quality of the living environment continue to rise, China is bound to increase capital investment in the construction of underground sewage treatment plants.

Overall, the construction of underground sewage treatment plants in China is booming at present, but the existing problems cannot be ignored. Finding ways to overcome and alleviate the above-mentioned problems through technological innovation, economic measures and policy support are key to determine the application and development prospects of underground sewage treatment plants in China in the future.

## 7. Comparative Analysis

The process, engineering, technology, cost, expenses, and existing problems of the underground, above-ground, closed and above-ground open sewage treatment plants are compared and analyzed, as shown in Table 6.

Based on the comparative analysis results in Table 6, it can be seen that underground and above-ground sewage treatment plants have their own advantages and disadvantages. From the perspective of saving construction investment and operating costs, a the above-ground construction form should be adopted; from the perspective of saving land, controlling odor, noise pollution, and beautifying the environment, the underground construction form should be selected.

**Table 6.** Comparisons of the underground, above-ground, closed and above-ground open sewage treatment plants.

| Project | Underground | Closed Above-Ground | Open Floor |
|---|---|---|---|
| Design form | The sewage plant and maintenance floor are underground and a park is built on the ground | The sewage facilities are built on the ground or are semi-underground, and the top of the pool is sealed with a cover | Above-ground or semi-underground, open cesspool |
| Land use | The structures are built together, compact, stacked up and down in layers, reducing the plane area and high land utilization rate | It occupies a large area and the land utilization rate is relatively low but it does not affect the surrounding environment; it will not affect the value of the surrounding land. | Occupies a large area, more than 30% of the green area; slightly affects the surrounding land value. |
| Noise pollution | The equipment is all underground, and mechanical noise and vibration will not affect the buildings and residents on the ground | Choosing low-noise equipment and closed equipment is conducive to noise reduction, which has little impact on the life and work of nearby residents | Takes noise reduction measures but cannot completely avoid the impact of noise on the life and work of the residents of the surrounding |
| Odor pollution | Fully enclosed underground management, comprehensive treatment of sewage, odor, and sludge, without impact on the environment and the lives of urban residents | The pool body is covered and sealed, and the odor is collected and processed in a centralized manner, and discharged after reaching the standard, eliminating the impact on the environment and the life of urban residents | The sewage plant is isolated from the surrounding environment through the green belt but it cannot completely avoid the impact on the life and work of the residents in the surrounding areas |
| Sealed form | The pool is stacked up and down in layers, reinforced concrete is sealed, multi-column network, and the structure is relatively complex | The top of the pool is covered, mostly with light-weight structures, such as fiberglass or color steel plates, which are relatively simple | No sealing cap |
| Construction difficulty | The underground depth is more than 10 m, requiring deep foundation pit support, and the construction is difficult | The buried depth is about 3 m underground, the excavation volume is small, and the construction difficulty is relatively small | The buried depth is about 3 m underground, the excavation volume is small, and the construction difficulty is relatively small |
| Construction investment | Invest CNY 5000–6000 per ton of water (full underground sealing and deodorization) | The investment per ton of water is about CNY 2500–3000 (covering and sealing and deodorization) | The investment per ton of water is about CNY 2000–2500 (without cover) |
| Floor area | 0.4~0.5 m$^2$/ton of water | 1~2 m$^2$/ton of water | 1~2 m$^2$/ton of water |
| Construction period | 12–18 months | 8–10 months | 6–8 months |
| Processing cost | Power consumption of 0.26 degrees/ton of water | Power consumption of 0.22 degrees/ton of water | Power consumption of 0.22 degrees/ton of water |

Extra care should be taken when choosing underground construction methods due to their high investment costs. The construction method for sewage treatment plants should be determined according to the current status of local urban development, economic level, and future development planning. Underground sewage treatment plants should not be built blindly. Underground or semi-underground non-sewage treatment plants can be considered in areas where land resources are relatively scarce or where there is a high requirement for the surrounding landscape of the plant. For some areas with relatively backward economy and rich land resources, the construction of above-ground sewage can still be reserved.

However, with the continuous development of China's economy, the rapid increase in the urbanization process and the increasing need for improvement in people's requirements for a better quality of living environment, the value of underground sewage treatment plants will become more obvious in the future.

## 8. Conclusions and Future Insights

1.  Underground sewage treatment plants have significant advantages in saving land occupation costs, pipeline network investment, in improving urban landscape and ecological environment as compared with above-ground sewage treatment plants. Their outstanding values can offset their relatively high construction and operating costs to a certain extent, thereby increasing their use in the future.

2.  The engineering design of an underground sewage treatment plant is different from that of a traditional above-ground sewage treatment plant. The engineering design of an underground sewage treatment plant should include the design of the treatment scale and degree of treatment, the design of the influent and effluent water quality, the site selection and form design of the project, the underground layout and structure design, the main process design, the single structure design, the ventilation and deodorization design, the lighting design, fire safety design, operation and maintenance design and the linkage design of ground landscape and underground sewage treatment plant construction.

3.  The above-ground landscape design of underground sewage treatment plants should pay attention to design principles, design features, water ecology, community friendliness and ecology friendliness, design style, and operation and maintenance costs. The establishment of an ecological complex can aid in the transformation of urban sewage treatment plants from negative assets to positive assets to meet the needs of future urban social and economic development and environmental protection.

4.  Innovation in design concepts, breakthroughs in key sewage treatment technologies, and diversification of investment and operation models can reduce the land occupation, investment, and operating costs of underground sewage treatment plants, and promote the realization of the social, economic and environmental values of underground sewage treatment plants. Moreover, large-scale semi-underground sewage treatment plants have broader development prospects in China than small-scale or fully underground sewage treatment plants.

**Author Contributions:** Conceptualization, A.S.G.; methodology, A.S.G.; writing—original draft preparation, writing—review and editing, N.A.; funding acquisition, A.S.G. All authors have read and agreed to the published version of the manuscript.

**Funding:** This work was supported by the Start-Up Funding for Research of Nanchang Institute of Science and Technology (NGRCZX-22-03), School of Environment and Civil Engineering, Nanchang, Jiangxi, China. This work was also supported by the Biomass and Wastewater Pollution Control Grant (no. 01160056) of the Green Intelligence Environmental School, Yangtze Normal University, and The China National Key Technology Support Program (no. 2014BAC27B01).

**Data Availability Statement:** No data supporting results.

**Conflicts of Interest:** The authors declare no conflict of interest.

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
