# Peer review of "An Extensive Analysis of the Engineering Design of Underground Sewage Plants in China"

_processes, doi:10.3390/pr11103010_

Round 1

Reviewer 1 Report

The paper is a review of technologies used in underground wastewater treatment systems. It is an interesting alternative to typical above-ground treatment plants. The material is of a popular science nature, but it is still worth publishing due to the lack of a collective description of this type of technology. These technologies are developmental and worth popularizing. Due to the review nature of the manuscript, it is difficult to make substantive comments. I draw the authors' attention to the abstract. It is not related to the content of the article, but contains some textbook information. It needs improvement.

Author Response

Reviewer_1:

The paper is a review of technologies used in underground wastewater treatment systems. It is an interesting alternative to typical above-ground treatment plants. The material is of a popular science nature, but it is still worth publishing due to the lack of a collective description of this type of technology. These technologies are developmental and worth popularizing. Due to the review nature of the manuscript, it is difficult to make substantive comments. I draw the authors' attention to the abstract. It is not related to the content of the article but contains some textbook information. It needs improvement.

Answer: On behalf of the co-author and me, we would like to express my gratitude to you for the opportunity to submit our revised manuscript for possible publication in the Process journal. We appreciate the encouraging response from the anonymous reviewer. Although it is a hot and significant topic to consider for writing a comprehensive review. We have revised our manuscript carefully based on the reviewers’ comments. We also wish to thank the anonymous reviewers for their observations, which we believe helped us to improve the quality and clarity of our review manuscript. The abstract was rewritten to explain the sewage plant design scenario in the country. Please refer to the abstract part. The English language was improved by a native speaker and an expert professor in the relevant field.

Remarks from Corresponding authors:

Thank you very much for your scrupulous review and insightful comments on our manuscript. We have revised the manuscript according to the comments, and carefully proofread the manuscript to minimize typographical, grammatical, and bibliographical errors and fine scientific fluency. We have requested two scientific editors from the United States the proofread our manuscript. They have reviewed the entire manuscript for a final check-up. The manuscript has been modified with reference to the reviewers’ comments. The changes are tracked with red marks.

Look forward to hearing from you. If the reviewer has more questions or needs more polish, we will be happy to do that.

Nasir Ali, (PhD)

Institute of Biotechnology Genetic Engineering, The University of Agriculture, Peshawar, 25130, Khyber Pakhtunkhwa, Pakistan

E-mail addresses: (N. Ali: [email protected])

The University of Agriculture, Peshawar, 25130, Khyber Pakhtunkhwa, Pakistan

Reviewer 2 Report

Dear authors,

thank you for your manuscript. The topic is basically interesting, but perhaps it is an early stage or work in progress?

Structure:

It seems that major parts of the manuscript have a kind of introductory style. Purpose of the paper is mentioned on page 13. Relation of the first sections to the overall topic is sometimes vague. In some passages you repeat text from former sections, in some there are argumentation loops. Clear statement about the contribution and the quality of conclusions is missing.

Contribution:

In the current state, I don't see the practical or scientific contribution of the manuscript. Regarding the first, you collect some data about existing plants and technologies but I don't feel that this is complete since you give me no idea about your methodological approach to collect the data. The data is not processed or related to the context where the plants are set in (catchment area, throughput, robustness, performance indicators, ...) so that it is not really possible to compare them.

The passages about engineering design are not valuable in the sense that a designer is supported in his or her activities as you do not show design principles, recommendations, guidelines or any modeling. It seems more a collection of basic layout classes (figure 6 which is a table).

Regarding the scientific contribution, you do not mention why the design of plants in China should be different from that in other countries or what requirements (not on an abstract but a distinct level) are posed. In some sentences you mention economic and social benefits. What do you mean by that? And why does it make sense to restrict basic parts of your literature work on Chinese sources? The review part of the paper should be delimited.

In your conclusions, please highlight at which place you go beyond the state of the art.

Language and form:

Extensive language review necessary, please try to avoid breaking up tables themselves or table and caption.

Please perform an extensive language editing, there are many typos, double spaces, ... in the text.

Author Response

Reviewer_2:

Thank you for your manuscript. The topic is basically interesting, but perhaps it is an early stage or a work in progress?

Answer: We are thankful to the anonymous reviewer and appreciate the comments for his deep interest in our manuscript. On behalf of the co-author and me, we would like to express my gratitude to you for the opportunity to submit our revised manuscript for possible publication in the Process journal. This manuscript is mainly focused on the underground sewage treatment plant which can remarkably reduce land occupation with less environmental pollution. We have already published several literatures on bio-natural gases production, working in Chinese educational institutes. China faces big challenges in the design of these sewage plants due to higher demands for biogases from the sewage plants in China. If the reviewer is more interested in the future of China biogas, please refer to this citation from the authors: (https://www.sciencedirect.com/science/article/pii/S2352484720313809). These are actually the early stages of planning to design underground plants to save agricultural land, reduce pollution, and consume low energy. We have mainly focused on the technical issues of underground sewage plants’ benefits over the above and other sewage plants. This is our own-oriented literature study to elaborate on the needs of concerned technology plantations in the country. 

Structure:

It seems that major parts of the manuscript have a kind of introductory style. Purpose of the paper is mentioned on page 13. Relation of the first sections to the overall topic is sometimes vague. In some passages you repeat text from former sections, in some there are argumentation loops. Clear statement about the contribution and the quality of conclusions is missing.

Answer: We appreciate critical comments from the reviewer in order to improve the overall manuscript structure. This manuscript has nothing with any textbooks. This is our own modulated insights about the sewage plants designed to reduce land occupation and pollution, as China has a huge population than the other countries. We have restructured the manuscript. The objectives section was moved after the introduction for a better understanding of the scope of the manuscript. The first introduction section was revised and improved with fluency. We have revised the overall manuscript to reduce the repetition and argumentation loop. The contribution, conclusion, and future endeavors are added in section 7, and the conclusion section. We have tried our best to address the comments from the anonymous reviewer. We really appreciate the deep interest of the reviewer in this manuscript. If the reviewer has more reservations, we will be happy to address them.

Contribution:

In the current state, I don't see the practical or scientific contribution of the manuscript. Regarding the first, you collect some data about existing plants and technologies but I don't feel that this is complete since you give me no idea about your methodological approach to collect the data.

Answer: We appreciate the deep critical comments from the reviewer and interest in our modulated study. The practical and scientific application of this manuscript is already discussed in page no. 41 with the following lines.

(1) As a new environment-friendly and resource-saving sewage treatment model, underground sewage treatment plants have significant advantages in saving land occupation costs and pipeline network investment compared with above-ground sewage treatment plants and have advantages in urban landscapes and ecological environments. Its outstanding value can offset its relatively high construction and operating costs to a certain extent, and it will be more widely used in the future.

(2) The engineering design of the underground sewage treatment plant is different from that of the traditional above-ground sewage treatment plant. The engineering design of an underground sewage treatment plant should include the design of the treatment scale and degree of treatment, the design of the influent and effluent water quality, the site selection and form design of the project, the underground layout and structure design, the main process design, the single structure design, the ventilation and deodorization design, the lighting design, fire safety design, operation and maintenance design and linkage design of ground landscape and underground sewage treatment plant construction, etc.

(3) The aboveground landscape design of underground sewage treatment plants should pay attention to design principles, design features, water ecology, combination with community friendliness and ecology friendliness, design style, and operation and maintenance costs; the establishment of an ecological complex can realize the transformation of urban sewage treatment plants from negative assets to positive assets meets the needs of future urban social and economic development and environmental protection.

(4) Innovation in design concepts, breakthroughs in key sewage treatment technologies, and diversification of investment and operation models can reduce the land occupation, investment, and operating costs of underground sewage treatment plants, and promote the realization of the social and economic value of underground sewage treatment plants and Environmental value, large-scale, semi-underground sewage treatment plants have broader development prospects in my country than small-scale or fully underground sewage treatment plants.

The methodological approach:

The complete methodology is not shown in the manuscript, as we just discussed the tactics of the underground treatment plant types. If the reviewer is interested in the methodology of the underground treatment plant, I have explained the completed methodology design.

Generally speaking, underground sewage treatment facilities can be divided into two types: underground sewage treatment plants and small underground sewage treatment stations. Among them, underground sewage treatment plants have two types: buried sewage treatment plants and cave-type (tunnel-type) sewage treatment plants. The former is the most common, while the latter is less used.

(1) Buried sewage treatment plant

An underground sewage treatment plant, also known as a fully enclosed underground sewage treatment plant, buries all structures or main treatment units into the ground, leaving only part of the testing room, maintenance room, control room, or other office units on the ground. At the same time, public facilities such as parks, parking lots, and residential buildings can be built above the ground to maximize the space utilization rate under a certain area. The technology of underground sewage treatment plants abroad has been going through a considerable period of time. For example, in Finland, Sweden, Czechoslovakia, and other countries, due to the relative shortage of land resources, sewage treatment plants often choose to be built underground.

(2) Cave-type (tunnel-type) sewage treatment plant

In some areas, affected by local geological and geomorphological conditions, sewage treatment plants are not suitable for underground construction but are more suitable for tunnel-type construction in the mountains. A typical case is the sewage treatment plant in Hayama Town, Kanagawa Prefecture, Japan. The sewage treatment plant adopts a mountain tunnel design model, and the main structures are in mountain tunnels. Compared with the above-ground sewage treatment plant, it occupies 1/3 of the land area of the above-ground sewage treatment plant. At the same time, since the main body of the sewage treatment plant is arranged in a tunnel, it has little impact on the surrounding environment and landscape, and there is no need for excessive greening and decoration. The Toulon sewage treatment Plant in France also adopts a tunnel-type design in the mountains. The water plant is built in a cave at the junction of the sea and the island. The treatment scale is 100,000 tons/day and the equivalent serving population is 550,000. The Stanley Sewage Treatment Plant in Hong Kong, built-in 1995 adopts a cave-style design model. All sewage treatment structures are in rock caves and have little environmental impact on the surrounding area. The layout of the internal structures of the water plant is shown in Fig. 1.

Fig. 1. Internal structure of Stanley Underground Sewage Treatment Plant in Hong Kong

In addition, the Henriksdal sewage treatment plant in Sweden will be expanded to become the world's largest cave-type underground sewage treatment plant. As shown in the figure Fig. 2, the sewage treatment plant adopts the A2O+MBR process and its processing capacity will reach 250,000 tons/day. The project is expected to be completed in 2028.

Fig. 2. Cave-type underground sewage treatment plant.

(3) Small underground sewage treatment plant

Small underground sewage treatment plants (stations) are different from conventional underground sewage treatment plants. In addition to the significant differences in the scale of treatment between the two, another important difference is that the above-ground parts of conventional underground sewage treatment plants can be reused, such as parks, stadiums, parking lots, etc.; And the upper parts of small underground sewage treatment facilities need to be equipped with some installation and access holes, observation holes and other passages, so the above-ground part of the land cannot be used for other purposes, and is generally only used for the construction of simple sewage treatment structures and greening construction. Small underground sewage treatment plants (stations) are mainly used to serve production or living areas with limited land resources, relatively concentrated residential areas, and long sewage collection distances from large sewage treatment plants, and the output of sewage treatment plants is generally less than 5,000 tons/day. While effectively ensuring the treatment effect, it increases the green area and reduces the impact of the stench on the living environment of surrounding residents.

At present, small underground sewage treatment plants (stations) are generally divided into three categories in the industry: underground integrated treatment equipment (complete sets of metal material equipment directly buried in the ground), septic tanks, and underground integrated treatment stations, which are used to meet the needs of the nearby treatment of domestic sewage of fewer than 5,000 tons/day in the region, saving investment in the pipe network. Among these three types of small underground sewage treatment stations, underground integrated treatment stations are widely used. Concrete is used as the material. The equipment room and the water treatment tank can be built and buried underground. It can be regarded as the prototype of an underground sewage treatment plant. Compared with the other two types, it has obvious advantages in terms of thermal insulation, anti-floating, and service life of the pool body, but the disadvantage lies in the large area, long construction time, and greater difficulty.

Regarding the scientific contribution, you do not mention why the design of plants in China should be different from that in other countries or what requirements. In some sentences, you mention economic and social benefits. What do you mean by that? And why does it make sense to restrict basic parts of your literature work to Chinese sources?

Answer: We are thankful to the reviewer for his deep interest in our manuscript and critical notes. As this modulated review manuscript is mostly focused on China USTP, so there is less information mentioned about abroad system engineering plants.

In view of the many problems in the construction and application of underground sewage treatment plants in China, this paper intends to conduct research on the design form and main process optimization selection of underground sewage treatment plants, engineering structure design, ground landscape design, etc. It also analyzes and discusses the key issues in the development and application of underground sewage treatment plants suitable for China's national conditions in the future, and provides technical support and a theoretical basis for the in-depth promotion and wide application of underground sewage treatment plants in China in the future. This article combines the literature research and the field research of some underground sewage treatment plants in operation in China. (Page no. 6). Economical benefits:

We determine the economic value of the STP as the total of land occupation cost, construction investment, operation and maintenance cost, and pipeline network investment. The planning land of the ASTP includes not only the construction land (15.00–49.10 ha), i.e., the land used for the sewage treatment structures, and the road inside, but also the greenbelt land (1.50–7.50 ha) which usually covers the surrounding area 200–300 metres away from the STP. The USTP needs much less land for construction. On the one hand, all the sewage treatment structures are arranged compactly based on integration and common-wall technology. On the other hand, no greenbelt (0.00 ha) is needed because all the noise and odor-producing units are covered and the gaseous pollutants are collected and treated underground. Taking the sewage treatment scale of the STPs into consideration, it can be found that the area covered for the treatment of per m3 sewage ranges from 0.57 m2/m3 to 0.83 m2/m3 in the ASTP, which is more than two times that of the USTP, changing between 0.23 m2/m3 and 0.29 m2/m3. What cannot be ignored is that, except for less land occupation for the construction of the USTP, another apparent advantage is the use of the above-ground space. Related data show that 98% of the total above-ground space of the USTP can be used for green land or the arrangement of communal facilities, and no more than 2% is needed to be used for functional units such as vents, central control rooms, and laboratories. Thus, it can be confirmed that the USTP will cover almost no land one day with the further development of the above-ground space. Please refer to sections 6 and 7.

Language and form:

Please perform an extensive language editing, there are many typos, double spaces.

Answer:

Remarks from Corresponding authors:

Thank you very much for your scrupulous review and insightful comments on our manuscript. We have revised the manuscript according to the comments, and carefully proofread the manuscript to minimize typographical, grammatical, and bibliographical errors and fine scientific fluency. We have requested two scientific editors from the United States the proofread our manuscript. They have reviewed the entire manuscript for a final check-up. The manuscript has been modified with reference to the reviewers’ comments. The changes are tracked with red marks.

Look forward to hearing from you. If the reviewer has more questions or needs more polish, we will be happy to do that.

Nasir Ali, (PhD)

Institute of Biotechnology Genetic Engineering, The University of Agriculture, Peshawar, 25130, Khyber Pakhtunkhwa, Pakistan

E-mail addresses: (N. Ali: [email protected])

The University of Agriculture, Peshawar, 25130, Khyber Pakhtunkhwa, Pakistan

Round 2

Reviewer 2 Report

Dear authors,

thank you for the revised manuscript. I put both versions side-by-side and have seen only moderate changes. Many issues that you raise in the answers to my comments are good to understand the background.

What I have seen as changes, beside the language (still not finalized? - double spaces, some spelling issues, section number in the conclusion missing, punctuation in tables, ...), are:

- A new abstract, which is a good improvement.

- A rearrangement of some parts of the manuscript (basically former section 6 is now 2) which from my point of view even weakens the manuscript as you now mention the basic processes after the applications.

The basic critisism regarding your research design is the same as before. I now understood that you want to compile the knowledge for the single underground sewage plant designs and see this as advancement compared to the state of the art.

I could agree to this, if you would do this on a general level, but not focussed on China (why this restriction?). The compilation of general design knowledge should be applicable independently from a location, shouldn't it? So my comment is to include worldwide applications of this technology in your abstraction of design knowledge and dive deeper into this design knowledge (processes, restrictions, operation, ...). Please guide the reader in a clearer and more rigorous way to the points (first talk about the processes and sewage in general, then make your study in China, then include a deeper study about the world wide applications (or vice versa), then compare the requirements, afterwards abstract design knowledge (deeper!)). The Table 6 contains many obvious points, please decompose them and take into account also the operation in a deeper way.

I'm very sorry, but following this I still would encourage you to improve the design of the manuscript and target the issue regarding the research design. In the current version with its minimal invasive changes I still vote for rejecting (with some tendencies to major revision) the paper in the current form for publication in Processes.

Improved but not completed.

Author Response

Dear authors,

Thank you for the revised manuscript. I put both versions side-by-side and have seen only moderate changes. Many issues that you raise in the answers to my comments are good to understand the background.

What I have seen as changes, besides the language (still not finalized? - double spaces, some spelling issues, section number in the conclusion missing), are:

- A new abstract, which is a good improvement.

Answer: Thank you for your encouraging comments. Indeed, we put the information for you to know the background and the samples taken from underground treatment plants. W also put our previously published paper in Energy report, to emphasize the need and necessity of the manuscript.

For double spaces and grammatical mistakes, we tried our best to improve. I think we belong to the Asian countries and 100 percent perfection is almost impossible for us. I guess the production team can do that minor improvement.

The section number for the conclusion part has been added. The abstract was rewritten to improve the overall structure of our manuscript for better understanding.

The basic critisism regarding your research design is the same as before. I now understood that you want to compile the knowledge for the single underground sewage plant designs and see this as an advancement compared to the state of the art.

I could agree to this, if you would do this on a general level, but not focussed on China (why this restriction?). The compilation of general design knowledge should be applicable independently from a location, shouldn't it? So my comment is to include worldwide applications of this technology in your abstraction of design knowledge and dive deeper into this design knowledge (processes, restrictions, operation, ...). Please guide the reader in a clearer and more rigorous way to the points (first talk about the processes and sewage in general, then make your study in China, then include a deeper study about the worldwide applications (or vice versa), then compare the requirements, afterward abstract design knowledge (deeper!)).

Answer: Thank you for your deep comment but I wrote it generally, not very specific. I have described our own modulated review article. I can make it general, not specific to China, if you are still interested. Me and my author did some research about the underground treatment plant in the past, so we decide to capture some information including the history, purpose and objectives, developmental processes, comparison, development, applications, problem and challenges, the comparative analysis regarding designs, conclusion, and future insights.

Again, if you are still interested, we can make a general review manuscript instead of focusing on China. This is what we know about the knowledge of single underground sewage plant designs and their advancement, future perspectives, and challenges.

Thank you for your time and comments. We appreciate that.

Round 3

Reviewer 2 Report

No more comments.

No more comments.